# Integration of OWL Password-Authenticated Key Exchange Protocol to Enhance IoT Application Protocols

**DOI:** 10.3390/s25082468

**Published:** 2025-04-14

**Authors:** Yair Rivera Julio, Angel Pinto Mangones, Juan Torres Tovio, María Clara Gómez-Álvarez, Dixon Salcedo

**Affiliations:** 1Department of Computer Science, Coporación Universitaria Americana, Barranquilla 08001, Colombia; 2Department of Computer Science, Universidad del Sinú, Montería 230001, Colombia; juantorrest@unisinu.edu.co; 3Departamento de Ciencias de la Computación y la Decisión, Facultad de Minas, Universidad Nacional Sede Medellín, Medellín P.O. Box 3840, Colombia; mcgomez@unal.edu.co; 4Computer Science and Electronic Department, Universidad de la Costa CUC, Barranquilla 080002, Colombia; dsalcedo2@cuc.edu.co

**Keywords:** internet of things (IoT), security protocols, one-message weak leakage-resilient PAKE (OWL), key-exchange protocol, man-in-the-middle (MitM), constrained application protocol (CoAP), message queuing telemetry transport (MQTT)

## Abstract

The rapid expansion of the IoT has led to increasing concerns about security, particularly in the early stages of communication where many IoT application-layer protocols, such as CoAP and MQTT, lack native support for secure key exchange. This absence exposes IoT systems to critical vulnerabilities, including dictionary attacks, session hijacking, and MitM threats, especially in resource-constrained environments. To address this challenge, this paper proposes the integration of OWL, a password-authenticated key exchange (PAKE) protocol, into existing IoT communication frameworks. OWL introduces a lightweight and secure mechanism for establishing high-entropy session keys from low-entropy credentials, without reliance on complex certificate infrastructures. Its one-round exchange model and resistance to both passive and active attacks make it particularly well-suited for constrained devices and dynamic network topologies. The originality of the proposal lies in embedding OWL directly into protocols like CoAP, enabling secure session establishment as a native feature rather than as an auxiliary security layer. Experimental results and formal analysis indicate that OWL achieves reduced authentication latency and lower computational overhead, while enhancing scalability, resilience, and protocol performance. The proposed solution provides an innovative, practical, and efficient framework for securing IoT communications from the foundational protocol level.

## 1. Introduction

The rapid proliferation of IoT technologies is transforming modern infrastructures across multiple domains, including smart homes, healthcare, industrial automation, and smart cities. This accelerated growth in connectivity fosters continuous data exchange between heterogeneous devices, enabling advanced automation, monitoring, and control capabilities. However, this dynamic and distributed ecosystem introduces complex challenges related to cybersecurity, privacy, and the integrity of communications, particularly due to the inherent resource constraints of IoT devices, which often operate with limited processing power, memory, and energy [1,2].

Although traditional security protocols remain robust in conventional computing environments, they frequently prove inadequate within IoT contexts. Their computational complexity, reliance on centralized certificate infrastructures, and configuration overhead hinder applicability in networks composed of low-power devices. Furthermore, the diversity of hardware and software platforms in IoT ecosystems complicates the deployment of uniform security solutions, thereby necessitating flexible, lightweight, and scalable approaches capable of adapting to varying operational requirements while maintaining strong guarantees of confidentiality, integrity, and authentication.

A particularly critical security gap is evident during the initial stages of communication, where key exchange and session establishment take place. Widely adopted IoT application-layer protocols, such as the Constrained Application Protocol (CoAP) and Message Queuing Telemetry Transport (MQTT), do not natively include secure key exchange mechanisms. Instead, these protocols rely on transport-layer solutions like Datagram Transport Layer Security (DTLS) or Transport Layer Security (TLS) to provide protection. Although effective in traditional networks, such frameworks often impose excessive computational load, increase handshake latency, and complicate certificate management in IoT deployments, ultimately degrading performance and exposing systems to threats such as dictionary attacks, replay attacks, and MitM intrusions, particularly in dynamic or intermittently connected environments.

To address key exchange limitations in constrained IoT environments, this work integrates OWL, a one-round Password-Authenticated Key Exchange (PAKE) protocol, into the application communication stack. The protocol enables high-entropy session key generation based on low-entropy credentials through lightweight cryptographic operations and zero-knowledge proofs (ZKPs). Its architecture eliminates reliance on certificate-based infrastructures, thereby reducing complexity and computational load. The one-message exchange structure minimizes handshake overhead and latency, making the protocol particularly well-suited for resource-constrained and intermittently connected IoT scenarios [3].

This work presents the integration of the OWL protocol within the CoAP stack to enable native support for secure key exchange in constrained environments while maintaining compatibility with existing IoT standards.This solution enhances security at the application layer, simplifies provisioning and session management, and reduces the attack surface during authentication phases. Furthermore, the proposed integration ensures forward secrecy and resilience against common attack vectors without incurring the computational burden typically associated with traditional methods.

This approach aims to establish a lightweight, scalable, and secure foundation for IoT communications. By embedding strong authentication and key exchange mechanisms directly within the application protocol, the OWL-CoAP integration transitions security from an auxiliary component to a core design principle, addressing a critical requirement in modern IoT architecture. Accordingly, the structure of this article is organized as follows: Section 2 provides a comprehensive review of existing IoT security protocols, highlighting their limitations and the evolution of cryptographic approaches such as PAKE schemes. Section 3 outlines the criteria for selecting suitable IoT protocols for OWL integration, analyzing trade-offs related to efficiency, scalability, and security. Section 4 details the architecture of the OWL-CoAP integration, describing its components, communication flow, and provisioning strategies. Section 5 explores how OWL enhances entropy generation and session key establishment within CoAP, supported by formal mathematical formulations and cryptographic techniques. Section 6 presents the experimental design, performance metrics, and comparative evaluation of OWL-CoAP against DTLS-X.509 and CoAP-PSK implementations. Section 7 discusses the implications of the results, deployment considerations, and outstanding challenges. Finally, Section 8 concludes the article by summarizing the main contributions and identifying potential future research directions to strengthen secure communication in IoT environments.

## 2. State of the Art

Security within the IoT ecosystem is undergoing a paradigm shift, driven by the incorporation of advanced cryptographic mechanisms that address emerging threats in critical domains such as remote healthcare and autonomous urban infrastructures. The continuous evolution and refinement of foundational protocols—namely, MQTT and CoAP—have been instrumental in this transformation, facilitating the integration of security as a core architectural element rather than as a subsequent enhancement. Notably, MQTT has demonstrated considerable progress by incorporating authentication and encryption enhancements essential in contexts where operational efficiency and robust security must coexist seamlessly. These security improvements not only fortify systems against unauthorized access and cyber threats but also foster greater confidence in the deployment of IoT technologies in applications critical to societal welfare [4].

MQTT’s advancements in IoT security epitomize the ongoing evolution of cryptographic practices tailored to increasingly complex and heterogeneous networks. Confronted with the challenge of securing vast and dispersed infrastructures, MQTT has adopted sophisticated cryptographic frameworks capable of accommodating the demands of modern IoT deployments. Furthermore, innovations in session management and token- or certificate-based authentication mechanisms have elevated the security posture of IoT applications. Collectively, these developments position MQTT not only as an efficient communication protocol but also as a secure conduit for transmitting sensitive data within mission-critical contexts [5].

In parallel, CoAP has emerged as a key player in securing large-scale IoT deployments through its integration with DTLS. A defining feature of these protocols is their adaptability, enabling deployment across diverse scenarios ranging from residential networks to complex industrial systems. Security strategies employed by protocols such as Zigbee (utilizing AES encryption and shared keys), LoRaWAN, and Sigfox (tailored for low-bandwidth environments) illustrate how application-specific constraints influence protocol design. These protocols reinforce not only data confidentiality but also the association and access control phases, thereby improving resistance to unauthorized network access [6].

Within the framework of Industry 4.0, the incorporation of built-in security mechanisms into long-range industrial sensor network protocols is particularly critical. Zigbee has augmented its architecture with dynamic key renewal capabilities, offering resilience against prolonged attacks while remaining adaptive to evolving threat landscapes. Conversely, LoRaWAN employs unique per-device application keys, facilitating personalized encryption schemes that enhance node-level security. These initiatives reflect a dual strategy—reactive and proactive—intended to maintain network robustness while ensuring long-term scalability and adaptability. Integrating these cryptographic features into the core protocol design highlights the necessity of a resilient and flexible security foundation essential for sustaining industrial IoT operations [7].

The convergence of communication technologies such as NB-IoT, CAT-LTE, and LTE-M marks a significant advancement in IoT security, especially in scenarios requiring expansive coverage and reliable connectivity. NB-IoT stands out as a solution optimized for wide-area, low-bandwidth applications, relying on 3GPP-standard encryption to maintain secure communications in cellular networks. This makes it particularly effective for deployments in public infrastructure monitoring and precision agriculture [8].

In contrast, CAT-LTE addresses the need for high data throughput while maintaining energy efficiency, meeting the security requirements of high-stakes applications such as automotive systems and telemedicine. LTE-M further extends the capabilities of cellular IoT by supporting mobility and voice services, making it suitable for applications like vehicular tracking and real-time logistics. Its design prioritizes low latency and uninterrupted communication, supported by native cryptographic standards that uphold the integrity of transmitted data [9].

Across these diverse protocol landscapes, security is consistently embedded as a foundational element rather than a retrofitted feature. The deployment of modern encryption algorithms and dynamic key management protocols serves as a bulwark against contemporary threats while promoting the scalability of IoT infrastructures. These protocol-level security enhancements demonstrate a strategic, forward-looking approach that ensures not only immediate protection but also adaptability to future vulnerabilities in an increasingly interconnected digital ecosystem [10,11].

At the forefront of secure IoT communication, innovative cryptographic constructs such as J-PAKE, OPAQUE, and OWL represent significant progress in the development of secure, efficient authentication mechanisms tailored to constrained environments. Designed to satisfy current requirements and remain adaptable to future demands, these protocols mark a paradigm shift in the design of lightweight yet secure key exchange frameworks [12].

J-PAKE and OPAQUE offer robust privacy-preserving authentication techniques. J-PAKE, leveraging ZKPs, eliminates the need for transmitting sensitive secret keys, thereby preserving confidentiality even in untrusted channels. OPAQUE reinforces credential protection during the authentication process, ensuring that sensitive information remains secure even under adversarial conditions [13].

Among these, OWL emerges as a highly promising protocol in the PAKE domain, offering mutual authentication based on low-entropy passwords, a critical requirement in low-resource devices. OWL distinguishes itself through its capacity to derive high-entropy keys from minimal password input, achieving secure authentication without incurring the high computational cost characteristic of asymmetric schemes such as RSA. This efficiency renders OWL particularly attractive for resource-limited IoT environments.

Building on these attributes, the subsequent comparative table presents a structured evaluation of principal PAKE protocols, outlining their technical strengths, limitations, and key features. Special emphasis is placed on metrics such as efficiency, scalability, and compatibility with elliptic curve cryptography (ECC), factors of heightened importance in resource-constrained IoT scenarios. This comprehensive overview is intended to serve as a strategic reference for selecting the most appropriate PAKE protocol in accordance with specific application requirements and security objectives. Refer to Table 1.

OWL provides an efficient alternative in critical scenarios by using a simpler, less demanding symmetric encryption protocol like PAKE, enhancing IoT protocol robustness, and leading new standards. Its implementation improves proactive defenses and adapts to new vulnerabilities, making these protocols central to the next generation of IoT systems, securing diverse environments, from smart cities to industrial setups. As shown in Table 2, OWL’s approach to key exchange stands out compared to other IoT security protocols, particularly those that lack native mechanisms for establishing secure sessions. By addressing these gaps, OWL enhances authentication, scalability, and overall resilience in constrained environments, providing a lightweight but robust security solution for modern IoT deployments.

## 3. Integration Criteria for IoT Protocols in the OWL Protocol

The selection of a suitable protocol for integration with OWL is guided by the evaluation of the following essential criteria:Operational Efficiency and Resource Requirements: Prefer a protocol that requires low computational complexity and is efficient in resource management, especially in IoT environments with devices with limited capabilities.Security and Encryption Handling: The protocol’s ability to integrate robust security measures without complex reconfigurations, using mechanisms such as TLS or DTLS to ensure the protection of transmitted data.Adaptability and Scalability: The protocol’s ability to handle a large number of devices and adapt to the expansion and dynamic needs of physical medium access and the IoT environment.

Now, the careful selection of protocols such as MQTT, CoAP, Zigbee, LoRa, BLE, Z-Wave, NB-IoT, LTE-M, and Sigfox for integration with advanced systems like OWL is extensively justified by their strategic adaptation to the specific demands of the IoT industry. These protocols not only stand out for their unique capabilities, but are also aligned with emerging trends and critical requirements for robustness, efficiency, and security in various industrial sectors. This alignment ensures that integration with systems like OWL is not only technically and functionally sound but also strategically advantageous, maximizing the return on investment and ensuring scalability and security in complex and evolving IoT environments.

### 3.1. MQTT

Inefficiencies in Integration: Although MQTT effectively manages real-time communications, its dependence on TCP/IP and the need to handle security in multiple layers through TLS may complicate its integration with systems like OWL that require flexibility in intermittent or dynamic networks. Furthermore, the centralized broker architecture can be a bottleneck and a single point of failure, which is not ideal in critical applications requiring high availability and distributed security [38].

### 3.2. CoAP

Challenges for Integration: CoAP does not natively define specific methods for session key exchange. Instead, CoAP can be configured to use transport-level security via DTLS or apply additional security layers such as digital certificates or preshared keys. DTLS supports various key exchange mechanisms, including RSA, ECC (Elliptic Curve Cryptography), and PSK (Pre-Shared Keys) [39].

### 3.3. Zigbee and LoRa

Redundancy and Complexity: The robust security implementation in Zigbee and LoRa, especially AES encryption and network management specific to mesh and LPWAN, may not require the advanced authentication features offered by OWL, leading to redundancy that could unnecessarily complicate the existing link-layer infrastructure without providing proportional benefits. This could result in additional complexity and resource overhead without clear benefits [40,41].

### 3.4. BLE and Z-Wave

Limited Security Enhancement: Since BLE and Z-Wave already have effective security mechanisms for their short-range applications and personal devices, integration with OWL may not provide significant improvements. BLE employs a pairing process that includes key exchange to establish an encrypted connection. This is performed through various pairing methods such as Just Works, Passkey Entry, and Out-of-Band (OOB). Regarding Z-Wave, in its Z-Wave S2 version, it includes measures to ensure that only verified and authenticated devices can join the network, using a process called DSK (Device Specific Key) to confirm the identity of the devices during inclusion [42,43].

### 3.5. NB-IoT and LTE-M

Redundancy in Security Layers: These protocols use robust security layers provided by cellular network operators, including the use of a SIM card for authentication, specifically designed to meet telecommunications standards. The integration of OWL could duplicate these security layers, adding complexity without a proportional increase in protection or functionality, which is inefficient, especially in terms of energy consumption and network management [44].

### 3.6. Sigfox

Communication Limitations: Sigfox’s architecture, oriented towards unidirectional or limited bidirectional transmission of small amounts of data, makes integration with OWL practically unfeasible. OWL, designed for secure and dynamic authentications, would require a richer bidirectional interaction that Sigfox simply cannot efficiently support [45].

The diversity in the design of IoT protocols reflects the specificity of each for certain operational environments, from energy efficiency to geographical extension. Ideally, OWL integration should enrich existing capabilities without overburdening systems with superfluous technical complexities. This evaluation indicates that, while several protocols could benefit from integration with OWL in theory, CoAP stands out as the most optimal candidate. CoAP not only has an architecture that effectively supports OWL’s signaling but also facilitates registration and efficient management of nodes through its middleware. This protocol, characterized by its lightweight and efficient nature, and its advanced security implementation through DTLS, positions it as an ideal complement to OWL, especially in IoT applications requiring agile and robust security management [45].

### 3.7. Complementarity of OWL and CoAP

Orientation towards IoT: Both OWL and CoAP are designed for IoT environments. OWL provides a secure authentication and key-establishing mechanism, while CoAP offers a lightweight application layer protocol for communication between IoT devices with limited resources. Enhanced security: Integrating OWL with CoAP could significantly enhance security in communication between IoT devices. OWL can securely handle key exchange and authentication, while CoAP, especially when combined with DTLS or OSCORE, ensures that data in transit is secured at the application or transport layer. Interoperability: Using OWL for authentication and key exchange and CoAP for regular communication between devices, the strengths of both protocols can be leveraged, thus maximizing interoperability within a diverse IoT environment.

### 3.8. Integration Considerations

Security Implementation: It is crucial to correctly implement security mechanisms in both protocols. For example, ensuring that the key exchange performed by OWL and data communication through CoAP are protected against external attacks. Resource Management: Although both protocols are designed to be efficient in resource usage, integration must carefully manage system resource consumption to avoid overloading IoT devices with limited capabilities. Configuration and Management: The configuration of the integration between OWL and CoAP should be managed in a way that simplifies both initial implementation and long-term management, including the update and renewal of the key and security certificate. Rigorous Testing: It is essential to conduct thorough testing to ensure that integration is not only secure but also stable and capable of handling anticipated use scenarios without failures.

## 4. Integration of the OWL Protocol with CoAP: Architecture, Criteria, and Secure Provisioning

The exponential growth of the IoT has led to significant challenges in designing security mechanisms that balance robust protection with the limited resources of constrained devices. OWL, a password-authenticated key-exchange (PAKE) protocol, addresses these challenges by establishing high-entropy session keys from low-entropy credentials. This section unifies two core aspects of OWL’s adoption in IoT contexts: the criteria for selecting and integrating IoT protocols with OWL, and the corresponding architecture for OWL-CoAP implementation with secure provisioning.

**Integration Criteria and Protocol Selection.** The following criteria guide the integration of OWL into IoT protocols:*Operational Efficiency and Resource Requirements:* Favor protocols that minimize computational overhead and are compatible with low-power or memory-constrained devices.*Security and Encryption Handling:* Ensure robust security features without extensive reconfiguration or overly complex certificate management. Lightweight solutions such as DTLS with pre-shared keys or raw public keys typically align with these demands.*Adaptability and Scalability:* Select protocols capable of handling large numbers of endpoints and capable of scaling in dynamic IoT environments.

Among the surveyed options (MQTT, Zigbee, LoRa, NB-IoT, and others), *CoAP emerged as the most suitable* for OWL integration. Its constrained but flexible nature, REST-oriented design, and compatibility with DTLS security align naturally with OWL’s lightweight authentication mechanism.

**Overall Architecture.** The OWL-CoAP integration employs an efficient key-exchange model within the CoAP ecosystem, reinforcing security without imposing excessive overhead. Table 3 summarizes the primary components in this architecture, highlighting their roles in provisioning, session management, protocol adaptation, and DTLS-based encryption.

**Consolidated Communication Flow.** Once OWL is integrated into a CoAP environment, four major stages define the secure communication workflow:*Initial Provisioning:* Devices receive core cryptographic material, such as OWL-derived public keys and Access Control Lists. Identifiers are assigned to manage authentication and resource permissions effectively.*Session Initiation:* A client starts a CoAP session by sending its identity and relevant parameters. OWL validates the client, deriving a fresh, high-entropy session key.*Secure Message Transmission:* All subsequent CoAP messages, encapsulating requests and responses, are protected using the newly established session key. The DTLS layer ensures confidentiality and integrity, while the OWL-based session keys mitigate identity spoofing.*Ongoing Session Management:* The session remains active as devices periodically renew keys or ACL entries. This continuous maintenance includes updating DTLS session states and OWL parameters when new devices join or existing devices leave.

By incorporating OWL into CoAP, IoT ecosystems can leverage a *lightweight yet robust* security model. OWL improves session authentication and key establishment without burdening constrained endpoints, while CoAP’s RESTful design and DTLS-based encryption safeguard data in transit. The synergy between these protocols simplifies rollouts across diverse IoT deployments where efficiency, scalability, and strong cryptographic guarantees are paramount.

## 5. Entropy Augmentation in CoAP: Enhancing Security Through Optimized Computational Efficiency

The OWL protocol introduces advanced mechanisms to enhance entropy within the CoAP ecosystem, optimizing the balance between robust security and computational efficiency. This integration is particularly crucial in IoT and resource-constrained environments, where devices operate with limited processing capabilities, but demand strong cryptographic protections to prevent unauthorized access and data breaches. In alignment with the lightweight nature of CoAP, OWL enhances its authentication and session initiation processes while minimizing computational overhead.

During the registration phase, OWL securely interacts with CoAP to establish initial trust. The user computes and transmits the derived values based on their credentials, while the server challenges the user with a zero-knowledge proof mechanism to confirm possession of the secret information without exposing it. This ensures that sensitive data remain confidential, even in constrained network conditions typical of CoAP-based deployments. In the log-in phase, the protocol verifies that the generated session key corresponds to the one derived during registration, allowing secure mutual authentication within the restricted CoAP message structure.

For password updates, the protocol incorporates enhanced security measures tailored to the lightweight requirements of CoAP. The process begins with the user initiating a password change request authenticated through current credentials. The server issues a new ZKPs-based challenge, ensuring that the user still possesses the original secret without transmitting it over the network. The user computes a new password-derived value locally, using the existing salt and cryptographic hash functions, and securely transmits it to the server. The server validates the new credentials against the existing parameters and, upon successful verification, updates its stored values while securely invalidating the old ones. This process ensures that password updates are resistant to interception, replay attacks, and computational inefficiencies [46,47].

Each of these phases, registration, login, and password updates, will be further detailed, with a focus on their interaction with CoAP and their reliance on cryptographic techniques such as hashing and ZKPs. The seamless integration of the OWL protocol with CoAP ensures a secure, efficient, and lightweight solution for authentication in constrained environments, reinforcing the integrity and confidentiality of user credentials.

### 5.1. OWL Registration Process with DTLS in CoAP

The OWL protocol, when integrated with DTLS, provides a robust framework for secure authentication in CoAP-based systems, particularly in IoT environments with constrained devices. During the registration process, OWL leverages cryptographic hash functions to derive intermediate values from weak passwords provided by users. The process begins by computing:(1)t=H(U∥w)modq
where *H* is a secure hash function, *U* is the user identifier, and *w* is the password. This is followed by the following computation:(2)π=H(t)modq
enhancing resistance to dictionary attacks. Finally, the verifier is securely stored on the server as:(3)$$T=g^{t}\mathrm{mod}p$$

These steps effectively increase the entropy of authentication parameters while maintaining compatibility with constrained devices, a critical feature of CoAP.

The interaction between the client and the server during the registration and session establishment is further secured by DTLS, which provides encryption for transmitted data and protects against eavesdropping and tampering. For example, the server generates an ephemeral private key $x_{3}$ and computes its corresponding public key:(4)$$X_{3}=g^{x_{3}}\mathrm{mod}p$$

To authenticate the client without exposing sensitive data, the server issues a Zero-Knowledge Proof (ZKP) challenge *c*, calculated as:(5)$$c=H(g,X_{3},U)\mathrm{mod}q$$

The client responds by calculating:(6)$$r=x_{3}+c\;\cdot \;t\mathrm{mod}q,$$
allowing both parties to verify the relationship:(7)$$g^{r}\equiv X_{3}\;\cdot \;g^{\left(c\;\cdot \;t\right)}\mathrm{mod}p$$

This process ensures mutual authentication while preserving the confidentiality of private keys. See Figure 1.

In addition, OWL employs ephemeral private keys ($x_{1},x_{2},x_{3},x_{4}$) and their corresponding public keys ($X_{1},X_{2},X_{3},X_{4}$) to establish secure session keys. ZKPs ($\pi_{1},\pi_{2},\pi_{3},\pi_{4}$) are used to verify the ownership of these keys without revealing their values. During the session key agreement phase, the value β is calculated using contributions from both the user and server keys:(8)$$\beta =f(x_{1},x_{2},X_{3},X_{4})\mathrm{mod}q$$
ensuring that the derived session key reflects the input of both parties. A ZKPs associated with β, denoted as(9)$$\Pi_{\beta }=ZKP\left(\beta \right)$$
validates its correctness without exposing private keys.

The integration of OWL with DTLS enhances CoAP by providing secure, high-entropy session keys derived from passwords and preconfigured materials. DTLS ensures the confidentiality and integrity of transmitted data, complementing OWL’s ability to prevent identity spoofing attacks. Moreover, the lightweight operations of OWL, optimized for multiplicative groups and elliptic curves, make it particularly suitable for real-time applications in the IoT.

The transcript of all messages exchanged during a session, secured by DTLS, guarantees consistency between the client and the server, reducing the risk of communication discrepancies. This log includes key parameters such as public keys, challenges, responses, and session details, providing a comprehensive record to verify session integrity.

### 5.2. OWL Login Process with DTLS in CoAP

The OWL protocol, combined with DTLS, ensures robust mutual authentication and secure session establishment within CoAP-based systems. This integration is particularly advantageous in IoT environments where constrained devices require efficient yet secure communication protocols. The following outlines the main steps of the log-in process.

#### 5.2.1. User Initialization

The user computes the following values:(10)t=H(U∥w)modq
where *U* is the username, and *w* is the password.(11)π=H(t)modq
enhances resistance to dictionary attacks.

Public keys are computed as:(12)$$X_{1}=g^{x_{1}}\mathrm{mod}p,\;X_{2}=g^{x_{2}}\mathrm{mod}p$$
based on ephemeral private keys $x_{1}$ and $x_{2}$.

The user generates ZKPs $\Pi_{1}$ and $\Pi_{2}$ to prove ownership of $x_{1}$ and $x_{2}$ without revealing them. The user sends $U,X_{1},X_{2},\Pi_{1},\Pi_{2}$ to the server, encrypted and secured using DTLS.

#### 5.2.2. Server Validation

The server decrypts the data using DTLS and verifies the received ZKPs $\Pi_{1}$ and $\Pi_{2}$, ensuring $X_{2}\neq 1$. The server generates its own ephemeral public keys:(13)$$X_{3}=g^{x_{3}}\mathrm{mod}p,\;X_{4}=g^{x_{4}}\mathrm{mod}p$$
based on its private keys $x_{3}$ and $x_{4}$. The server produces ZKPs $\Pi_{3}$ and $\Pi_{4}$ to prove ownership of $x_{3}$ and $x_{4}$.

The server calculates a shared value for session key establishment:(14)$$\beta ={\left(X_{1}\;\cdot \;X_{2}\;\cdot \;X_{3}\right)}^{x_{4}\;\cdot \;\pi }\mathrm{mod}q$$
and generates $\Pi_{\beta }$, a ZKPs to validate β.

#### 5.2.3. Server Sends Validation Data

The server sends $S,X_{3},X_{4},\Pi_{3},\Pi_{4},\Pi_{\beta },\beta$ to the user over a secure DTLS channel.

#### 5.2.4. User Validation and Key Computation

The user verifies $\Pi_{3}$ and $\Pi_{4}$, ensuring $X_{4}\neq 1$. The user computes(15)$$\alpha ={\left(X_{1}\;\cdot \;X_{3}\;\cdot \;X_{4}\right)}^{x_{2}\;\cdot \;\pi }\mathrm{mod}q$$(16)$$K={\left(\beta /X_{2}^{x_{2}\;\cdot \;\pi }\right)}^{x_{2}}\mathrm{mod}q$$
as the session key.(17)h=H(K∥Transcript)
where the transcript includes all messages exchanged.(18)$$r=x_{1}-t\;\cdot \;h\mathrm{mod}q$$
a response value based on the session hash.

The user generates $\Pi_{\alpha }$, a ZKPs to prove the correctness of α, and sends $\alpha ,\Pi_{\alpha },r$ to the server through DTLS.

#### 5.2.5. Server Key Validation and Session Establishment

The server verifies $\Pi_{\alpha }$ and computes(19)$$K={\left(\beta /X_{4}^{x_{4}\;\cdot \;\pi }\right)}^{x_{4}}\mathrm{mod}q$$
as the session key.(20)h=H(K∥Transcript)The server verifies(21)$$g^{r}\;\cdot \;T^{h}\equiv X_{1},$$
confirming the user response.

Both the user and the server compute(22)k=H(K),
the final shared session key.

The integration of DTLS into the OWL login process ensures encrypted message exchanges, further enhancing confidentiality and integrity. See Figure 2.

### 5.3. Password Update in the OWL Protocol with DTLS Integration

The OWL protocol provides a secure and efficient mechanism for updating user passwords while preserving the integrity of the authentication process. When integrated with DTLS in CoAP-based systems, the password update process ensures that sensitive information remains encrypted and protected throughout the update, adding an additional layer of security.

#### 5.3.1. Password Update Process


**User Operations:**


The user selects a new password $w^{\prime }$. Using cryptographic hash functions, the user computes(23)$$t^{\prime }=H\left(U∥w^{\prime }\right)\mathrm{mod}q$$
where *H* is a secure hash function and *U* is the user’s identifier.(24)$$\pi^{\prime }=H\left(t^{\prime }\right)\mathrm{mod}q$$
enhances protection against dictionary attacks.

The user computes the verification value(25)$$T^{\prime }=g^{t^{\prime }}\mathrm{mod}p,$$
which will be used for future authentication attempts.

The user securely transmits $\pi^{\prime }$ and $T^{\prime }$ to the server over a DTLS-encrypted channel.

**Server Preparation:** The server decrypts and verifies the received $\pi^{\prime }$ and $T^{\prime }$ using DTLS to ensure integrity and authenticity. The server then replaces the existing values π and *T* with $\pi^{\prime }$ and $T^{\prime }$, securely storing the updated credentials for future authentication (see Figure 3).

#### 5.3.2. Security Implications

The password update process, reinforced with DTLS, ensures that:Sensitive information, such as the new password $w^{\prime }$, is never transmitted or stored in plaintext.All computations are performed using cryptographic hash functions, enhancing resistance to attacks such as eavesdropping or replay attacks.The updated values $\pi^{\prime }$ and $\;T^{\prime }$ are seamlessly integrated into the existing authentication framework, maintaining compatibility with the security guarantees of the protocol.DTLS encrypts the communication channel, ensuring confidentiality and integrity during the password update process.

This mechanism highlights the flexibility and robustness of the OWL protocol in adapting to dynamic credential updates, making it a secure solution for resource-constrained IoT devices in CoAP-based environments.

## 6. Formal Security Analysis of the OWL–CoAP–DTLS Protocol

In this section, a *game-based* security proof is presented for the proposed OWL–CoAP–DTLS protocol. The aim is to demonstrate that, under standard cryptographic assumptions (e.g., hardness of discrete logarithm, CDH / DDH, random oracle models for hash functions, and the soundness of ZKPs), an adversary’s advantage in breaking session key confidentiality or impersonating a legitimate party is negligible.

### 6.1. Security Model and Definitions

**Participants.** Two honest parties are considered, a client *C* and a server *S*, communicating over an insecure channel. They aim to run the OWL-CoAP-DTLS protocol to authenticate each other and derive a shared session key *k*.**Adversary A.** The adversary can intercept, modify, and forward messages freely (Dolev–Yao model). It may attempt to corrupt the credentials of one endpoint. The primary goals of A are:
**Key indistinguishability:** Break the confidentiality of the session key *k* or distinguish ciphertexts from random strings.**Impersonation:** Convince one party to accept a session key without the genuine participation of the other party, or impersonate one participant without possessing the correct password/credentials.**Security Game.** In accordance with the structure of *bit-guessing* or *IND-security* games, an experiment is defined between a *Challenger* and an adversary A. The Challenger emulates an honest execution of the protocol for both client and server roles, while A is allowed to interact with the system, intercept communications, and issue queries to various oracles. At the conclusion of the experiment, A outputs a guess $\hat{b}\in \left\{0,1\right\}$, indicating whether the challenge ciphertext or key token corresponds to a real instance (b=1) or to a randomly generated value (b=0). The *advantage* of A is quantified as$$\mathrm{Adv}\left(A\right)\;=\;\left|\mathrm{Pr}\left[\hat{b}=b\right]\;-\;\frac{1}{2}\right|.$$**Desired Result.** The objective is to demonstrate that for any polynomial-time adversary A, the advantage Adv(A) remains negligible under standard cryptographic assumptions.

### 6.2. Overview of the Game-Based Proof

The argument is developed through a sequence of games, beginning with the *Real Execution Game* (Game 0) and incrementally transformed into a final game in which the adversary’s success probability is either negligible or trivial. Each transition relies either on a well-known cryptographic assumption (e.g., CDH/DDH) or on the random oracle property of hash functions and ZKPs.

#### 6.2.1. Game 0: Real Protocol Execution

**Game 0** is the actual execution of the OWL-CoAP-DTLS protocol:The challenger runs OWL for the client and the server, generates ephemeral keys $x_{1},x_{2},x_{3},x_{4}$, computes public values $X_{1}=g^{x_{1}}\mathrm{mod}p$, etc., and performs the required ZKPs steps.A shared session key k=H(K) is established once zero-knowledge authentication and password-derived elements have been verified.The adversary A can observe, modify, or inject messages at will and ends by guessing the bit *b* in a challenge or attempting to impersonate a party.

Denote ${\mathrm{Adv}}^{\left(0\right)}\left(A\right)=\left|\mathrm{Pr}\left[\hat{b}=b\right]-\frac{1}{2}\right|$ as the adversary’s advantage in Game 0.

#### 6.2.2. Game 1: Randomization of Ephemeral Elements

In **Game 1**, the generation of ephemeral keys and responses in the zero-knowledge proofs (ZKPs) is modified in a manner that remains computationally indistinguishable to the adversary A, under the Computational Diffie–Hellman (CDH) or Decisional Diffie–Hellman (DDH) assumptions. Specifically:Rather than computing $X_{1}=g^{x_{1}}$, the value $X_{1}$ is replaced with a uniformly random element ${\tilde{X}}_{1}\in G$ in cases where such substitution is indistinguishable under the DDH assumption.Commitments and responses in ZKPs are simulated using random values that are consistent with the expected protocol transcript, assuming the presence of random oracles for hash-based commitments.

Standard reductions show that $\left|{\mathrm{Adv}}^{\left(1\right)}\left(A\right)-{\mathrm{Adv}}^{\left(0\right)}\left(A\right)\right|\leq \varepsilon_{1}$, where $\varepsilon_{1}$ is negligible.

#### 6.2.3. Game 2: Replacement of the Session Key with a Random Value

**Game 2** is the key step:Instead of deriving $k=H\left(f(x_{1},x_{2},X_{3},X_{4})\right)$ from real group operations, the challenger picks *k* uniformly at random from the key space.If the adversary could distinguish real key messages from random, it would imply the ability to solve the underlying hard problem (CDH, discrete log, or break the random oracle).

It follows that $\left|{\mathrm{Adv}}^{\left(2\right)}\left(A\right)-{\mathrm{Adv}}^{\left(1\right)}\left(A\right)\right|\leq \varepsilon_{2}$, where $\varepsilon_{2}$ is negligible under the assumed hardness of the underlying computational problem.

#### 6.2.4. Game 3: Trivial Response

Finally, in **Game 3**, the session key *k* is treated as an independent random value, and all the outputs of the protocol (e.g., DTLS records or final acceptance messages) are consistently simulated with a random key. At this point, the adversary has no better strategy than random guessing:$${\mathrm{Adv}}^{\left(3\right)}\left(A\right)\;=\;0\;\left(\mathrm{or}\;\mathrm{statistically}\;\mathrm{small}\right).$$By chaining these inequalities, it follows that$${\mathrm{Adv}}^{\left(0\right)}\left(A\right)\;\leq \;{\mathrm{Adv}}^{\left(1\right)}\left(A\right)+\varepsilon_{1}\;\approx \;{\mathrm{Adv}}^{\left(2\right)}\left(A\right)+\varepsilon_{1}+\varepsilon_{2}\;\approx \;{\mathrm{Adv}}^{\left(3\right)}\left(A\right)+\varepsilon_{1}+\varepsilon_{2}+\varepsilon_{3},$$
where $\varepsilon_{1},\varepsilon_{2},\varepsilon_{3}$ are all negligible. Consequently, the advantage of any adversary A is also negligible, thereby establishing security.

### 6.3. Discussion and Practical Considerations

**Resistance to Dictionary Attacks.** The protocol never exposes the low-entropy password directly. Instead, it uses a salted hash t=H(U∥w) and a verifier $T=g^{t}\mathrm{mod}p$. The ZKPs ensure an adversary cannot passively or actively guess the password without being detected.**Zero-Knowledge Commitments.** The underlying ZKPs steps rely on cryptographic hashing assumed as a random oracle, preventing an adversary from fabricating valid proofs without the genuine private exponent(s).**DTLS Layer.** Even if an attacker injects packets, the final key *k* remains indistinguishable from a random string. DTLS complements the PAKE layer by protecting subsequent messages once the handshake concludes.**Implementation Caveats.** This game-based proof assumes an idealized setting. Real-world deployments must address side-channel attacks (timing, power consumption), secure credential storage, reliable firmware updates, and robust random-number generation to preserve theoretical security guarantees.

Under standard cryptographic assumptions, the bit-guessing advantage of any adversary attempting to compromise the OWL-CoAP-DTLS protocol remains comparable to random guessing (i.e., 1/2). Consequently, the proposed method achieves both robust PAKE security and efficient session establishment for resource-constrained IoT environments while adhering to modern cryptographic practices.

## 7. Experimental Design

To rigorously evaluate the performance of the **OWL–CoAP integration** proposed within IoT environments, a **comprehensive experimental design** was developed to compare it with **traditional security approaches**. The evaluation focused on four key performance metrics: **computational efficiency, communication overhead, authentication latency, and security robustness**. These metrics were analyzed at multiple levels to account for varying operational conditions and device capabilities.

To ensure the **validity and reliability** of the results, several critical statistical considerations were incorporated into the experimental design, including **normality, randomization, and homoscedasticity** (homogeneity of variances). The assumption of normality was verified to confirm that the collected data followed a **Gaussian distribution**, ensuring the appropriateness of parametric statistical tests. Randomization was applied throughout the experiment to minimize bias and external influences, ensuring a fair comparison between **OWL–CoAP and conventional protocols**. Furthermore, homoscedasticity was assessed to confirm that variance levels remained consistent between different experimental groups, enhancing the **statistical significance** of the observed differences.

These methodological safeguards were implemented to provide a **robust assessment** of the performance differences between the evaluated protocols. **As detailed in Table 4, the study examines the impact of key factors such as key size, encryption methods, protocol types, and attack resilience on the selected performance metrics. This structured approach enables a comprehensive comparison of OWL–CoAP with alternative security mechanisms, offering valuable insights into its effectiveness and adaptability for various IoT deployment scenarios**.

### 7.1. Experimental Setup

The experimental environment consists of the following elements:**Software Environment:** The simulation is implemented in **Python** using the aiocoap library for CoAP communication.**Security Configuration:** The server operates over **DTLS on port 5684**, supporting both **unidirectional and bidirectional DTLS authentication**.**Network Simulation:** Network conditions, including **latency, bandwidth, and packet loss**, are emulated using Mininet and Linux Traffic Control (tc).**Evaluation Tools:** Data collection and monitoring are performed using **Wireshark**, custom Python scripts, and statistical analysis in **Matplotlib and Pandas**.**ThingsBoard Integration:** The simulated IoT devices securely communicate with a ThingsBoard server running a **CoAP DTLS endpoint**.

#### 7.1.1. Additional Simulation Environment Details

To enhance clarity and support reproducibility, the experiments were conducted under a controlled network topology configured via Mininet, emulating a **peer-to-peer (P2P) architecture** with **DTLS**-based communication channels to assess direct device-to-device interactions. This setup mirrors realistic IoT deployments where secure data exchange between nodes is crucial.

The network conditions were carefully modeled to reflect both ideal and constrained environments. Latency values ranged from 10 ms to 100 ms, while packet loss rates varied between 0.5% and 5%, managed through Linux Traffic Control (tc). These ranges were selected to represent conditions typical of smart building deployments, industrial IoT applications, and constrained wireless sensor networks (WSNs).

The DTLS configuration was adapted to simulate varying encryption complexities. Key sizes were tested from 1 bit up to 500 bits (at increments of 1 bit), aligning with typical ECC and password-derived key lengths explored in research on resource-constrained environments. The selected encryption methods included **DTLS-PSK**, **DTLS-X.509**, and **DTLS-OWL**, ensuring a comprehensive evaluation of lightweight, certificate-based, and optimized key exchange protocols.

Hash functions such as **SHA-256** were employed for message integrity, while encryption relied on **AES-GCM** with a 128-bit key for secure data transmission. This configuration follows best practices for IoT security frameworks and ensures alignment with real-world deployment standards.

This methodological rigor in selecting network parameters, cryptographic primitives, and secure P2P architecture strengthens both the internal validity of the experimental outcomes and their broader applicability to real-world IoT deployments. The incorporation of DTLS protocols further ensures that the presented results accurately reflect the latency, overhead, and security performance expected in secure IoT ecosystems.

#### 7.1.2. Security Configuration and Certificate Management

To enable secure CoAP communication with **DTLS**, the ThingsBoard instance is configured with **valid ECDSA certificates** stored in PEM format. The following configurations were applied:


export COAP_DTLS_ENABLED=true

export COAP_DTLS_CREDENTIALS_TYPE=PEM

export COAP_DTLS_PEM_CERT=server.pem

export COAP_DTLS_PEM_KEY=server_key.pem

export COAP_DTLS_PEM_KEY_PASSWORD=secret

export COAP_DTLS_BIND_PORT=5684


For testing purposes, self-signed ECDSA certificates were generated using the following commands:


openssl ecparam -out server_key.pem -name secp256r1 -genkey

openssl req -new -key server_key.pem -x509 -nodes -days 365 -out server.pem -subj "/CN=localhost"
**Note:** While self-signed certificates are useful for testing, it is recommended to use a **certificate from a trusted Certificate Authority (CA) for production deployments**.

### 7.2. Authentication Latency

Authentication latency is a critical factor in IoT security. In this study, a comparative analysis was performed on the authentication latency of the **CoAP-PSK, CoAP-X.509, and CoAP-OWL** protocols. The latency data were generated through simulations using a **normal distribution** with predefined mean and standard deviation values for each protocol.

Figure 4 presents the histogram of authentication latency, highlighting significant differences in performance between the evaluated protocols. **CoAP–OWL demonstrates the lowest average latency**, making it the most efficient solution for **IoT environments that require rapid authentication**. In contrast, **CoAP–X.509 exhibits the highest latency**, reflecting the additional processing time associated with **certificate-based authentication**. **CoAP–PSK falls between the two**, offering a **moderate trade-off between speed and security**.

Figure 5 provides a box plot visualization of latency dispersion between protocols. CoAP–OWL shows minimal variation, indicating consistent authentication times, while CoAP–X.509 reveals significant variability, attributed to complex cryptographic operations. CoAP–PSK exhibits moderate dispersion, suggesting stable but slightly variable authentication times compared to CoAP–OWL.

Table 5 summarizes the statistical results, including the mean latency and standard deviation for each protocol. CoAP–OWL achieves the best performance, with an average latency of 78.06 ms and the lowest standard deviation, reinforcing its suitability for time-sensitive IoT applications. CoAP–PSK, with a mean latency of 120.29 ms, provides a reasonable balance, whereas CoAP–X.509 incurs a significant overhead with an average latency of 181.77 ms.

The results show that CoAP–OWL not only offers lower latency but also provides more predictable and reliable authentication. Its lightweight design and efficient cryptographic approach make it an optimal choice for IoT deployments where responsiveness and resource efficiency are paramount. The CoAP–PSK protocol presents a viable alternative for scenarios where moderate security and authentication speed are required. In contrast, CoAP–X.509, while offering robust security, may not be suitable for applications with stringent latency requirements due to its computational complexity.

These findings underscore the importance of selecting an appropriate authentication mechanism based on the specific needs of the IoT application. The CoAP–OWL protocol stands out as the most effective solution to achieve secure and efficient communication in constrained environments, contributing to the overall resilience and performance of IoT ecosystems.

## 8. Computational Overhead

To evaluate the computational overhead of the OWL-DTLS and DTLS-X.509 protocols, a simulation was performed that focused on three critical metrics: computational complexity, energy consumption, and network overhead. The results provide insights into the suitability of these protocols for resource-constrained IoT environments by analyzing their performance across different key sizes, ranging from 1 to 500 bits, with 1000 iterations per key size to ensure statistical significance. The complexity models were derived considering key factors such as computational efficiency, security mechanisms, and protocol design.

The following is a comparison of the computational complexity between the **DTLS–X.509** and **OWL–DTLS** protocols, highlighting the advantages that OWL–DTLS offers in terms of performance and security. Although DTLS–X.509 relies on asymmetric cryptography with high computational cost, OWL–DTLS adopts more efficient techniques, such as PAKE, which optimizes authentication and session establishment.

Table 6 presents a detailed comparison of both protocols, illustrating key differences in terms of key generation, verification, handshake complexity, encryption overhead, and attack resilience.

In general, OWL–DTLS offers a significant reduction in computational complexity, resulting in lower resource consumption and shorter processing times. Optimization of verification and handshake operations contributes to faster and more efficient authentication, while its design, based on robust passwords, provides better resilience against attacks, overcoming the limitations of DTLS–X.509 in environments with high security and efficiency requirements. The simulation results for computational cost, illustrated in Figure 6, demonstrate that OWL–DTLS consistently outperforms DTLS–X.509 by reducing complexity, making it more suitable for low-resource devices. As summarized in Table 6, OWL–DTLS achieves this efficiency by reducing the complexity of key generation, minimizing handshake overhead, and eliminating the need for certificate validation, making it a more practical solution for constrained IoT environments.

In terms of energy consumption, Figure 7 indicates that OWL–DTLS consumes around 20% less energy compared to DTLS–X.509 across all key sizes, mainly due to the simplified cryptographic operations and reduced handshake complexity. The base energy consumption for OWL–DTLS is calculated at 0.05n Joules, whereas DTLS–X.509 incurs additional penalties due to the certificate verification process.

Regarding network overhead, the analysis presented in Figure 8 demonstrates that OWL–DTLS introduces significantly lower overhead, with an average reduction of 18% compared to DTLS–X.509. The improved efficiency of OWL–DTLS allows for reduced message sizes and fewer data exchanges, which is crucial in IoT scenarios where bandwidth is a limiting factor.

Table 7 summarizes a sample of the simulation results for both protocols:

The findings confirm that OWL–DTLS is a more efficient alternative for IoT applications due to its lower computational complexity, reduced energy consumption, and minimal network overhead. This makes OWL–DTLS an attractive solution for enhancing the security of constrained IoT environments while ensuring optimal performance.

## 9. Discussion

The experimental results provide a detailed perspective on the computational efficiency and communication performance of OWL–DTLS compared to DTLS–X.509. The data collected from the simulations demonstrate a clear reduction in computational overhead, energy consumption, and network resource usage, which are critical factors for IoT deployments in constrained environments.

The computational complexity analysis, shown in Figure 6, indicates that OWL–DTLS requires significantly fewer processing resources compared to DTLS–X.509. The simplified cryptographic operations and the use of PAKE-based authentication mechanisms contribute to reducing the overall processing burden. These reductions are particularly evident as key sizes increase, demonstrating the ability of OWL–DTLS to scale efficiently without introducing excessive computational demands.

Energy consumption is another critical aspect addressed in the experimental evaluation. As illustrated in Figure 7, OWL–DTLS consistently consumes less power compared to DTLS–X.509. This efficiency is attributed to the streamlined key exchange and authentication processes, which reduce the need for repetitive cryptographic operations. Lower energy requirements make OWL–DTLS a viable solution for battery-powered IoT devices operating in remote or resource-constrained environments.

The evaluation of network overhead, depicted in Figure 8, highlights the efficiency of OWL–DTLS in minimizing data transmission overhead. The results indicate an approximate 18% reduction in communication overhead compared to DTLS-X.509. This improvement stems from OWL–DTLS’s ability to optimize message exchanges and reduce redundant data transmissions, which is particularly beneficial in low-bandwidth IoT networks.

Despite these positive observations, several challenges persist. One of the primary concerns is the scalability of OWL–DTLS in large-scale IoT networks. As the number of connected devices increases, the management of authentication sessions and key distribution becomes increasingly complex. Efficient handling of these elements is crucial to maintaining performance and avoiding potential bottlenecks in high-density deployments.

Another area of interest is the adaptability of OWL–DTLS to different IoT communication protocols and network architectures. Given the diversity of IoT ecosystems, interoperability with existing protocols such as MQTT, CoAP, and LoRaWAN remains an ongoing area of exploration. Ensuring seamless integration without compromising security or performance is a key consideration for practical implementation.

Furthermore, while OWL–DTLS exhibits lower processing and energy costs, the trade-offs between security and performance need to be continuously assessed. Ensuring that reduced complexity does not compromise the robustness of the authentication process is essential to maintain the integrity of IoT communications, particularly in applications with stringent security requirements.

In general, the analysis of the experimental data underscores the potential advantages and areas for further optimization in the deployment of OWL–DTLS for IoT environments. Future evaluations should focus on dynamic testing scenarios, varying network loads, and real-world deployments to further validate the performance metrics observed in controlled simulations.

## 10. Conclusions

This work presents the integration of OWL into CoAP-based IoT systems secured with DTLS, offering a lightweight and robust solution for secure session establishment in resource-constrained environments. The OWL one-message key exchange, grounded in ZKPs and high-entropy session derivation, shows significant gains in computational efficiency, authentication latency, and network overhead compared to traditional certificate-based approaches such as DTLS–X.509.

Beyond performance, the proposed OWL–CoAP–DTLS architecture addresses critical security needs at the application layer, offering native support for secure authentication without reliance on complex certificate infrastructures. However, real-world deployment requires further considerations. Firmware update mechanisms must be secured to preserve the integrity of OWL credentials, and provisioning strategies must support dynamic key refresh without disrupting active sessions. Interoperability also remains a challenge: While OWL aligns naturally with CoAP, adapting its mechanisms to broker-based architectures like MQTT will require novel handshake coordination models that preserve lightweight design while ensuring mutual authentication.

In addition, deployment in heterogeneous environments must address device compatibility, protocol fragmentation, and session synchronization across dynamic topologies. Trade-offs may arise between strong security guarantees and real-time responsiveness, especially in mobile or lossy networks. Future work should explore hybrid deployment models, automated key management, and OWL integration into alternative IoT stacks to support broader adoption in industrial, healthcare, and smart city scenarios.

The proposed architecture not only enhances the confidentiality and integrity of IoT communications but also reduces the operational burden on constrained devices by minimizing handshake steps, simplifying key management, and avoiding certificate verification. This simplification enables scalable, energy-efficient, and low-latency secure communication, making OWL–CoAP–DTLS particularly suitable for dense IoT networks and environments with intermittent connectivity. By embedding security as a native component of the protocol stack, this approach redefines how trust is established in IoT ecosystems—shifting from heavyweight infrastructure-dependent models to lightweight, password-based cryptographic foundations that are easier to deploy, maintain, and scale. 

## Figures and Tables

**Figure 1 sensors-25-02468-f001:**
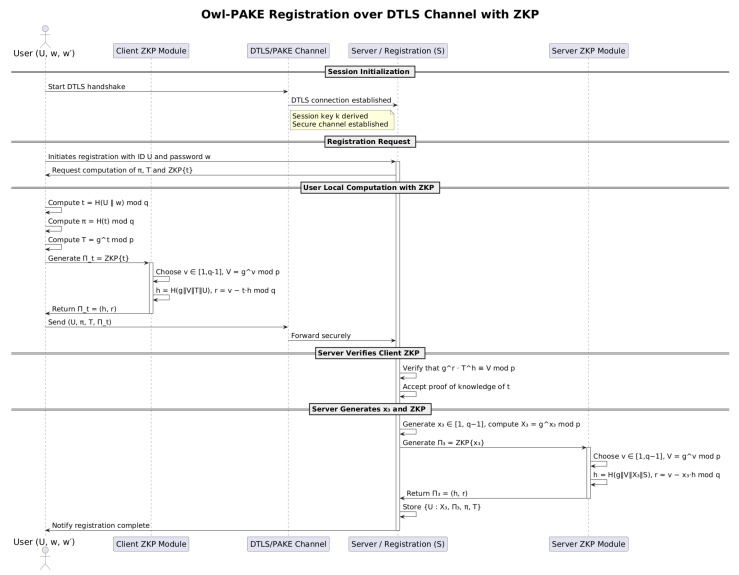
Registration in the OWL protocol using ZKPs.

**Figure 2 sensors-25-02468-f002:**
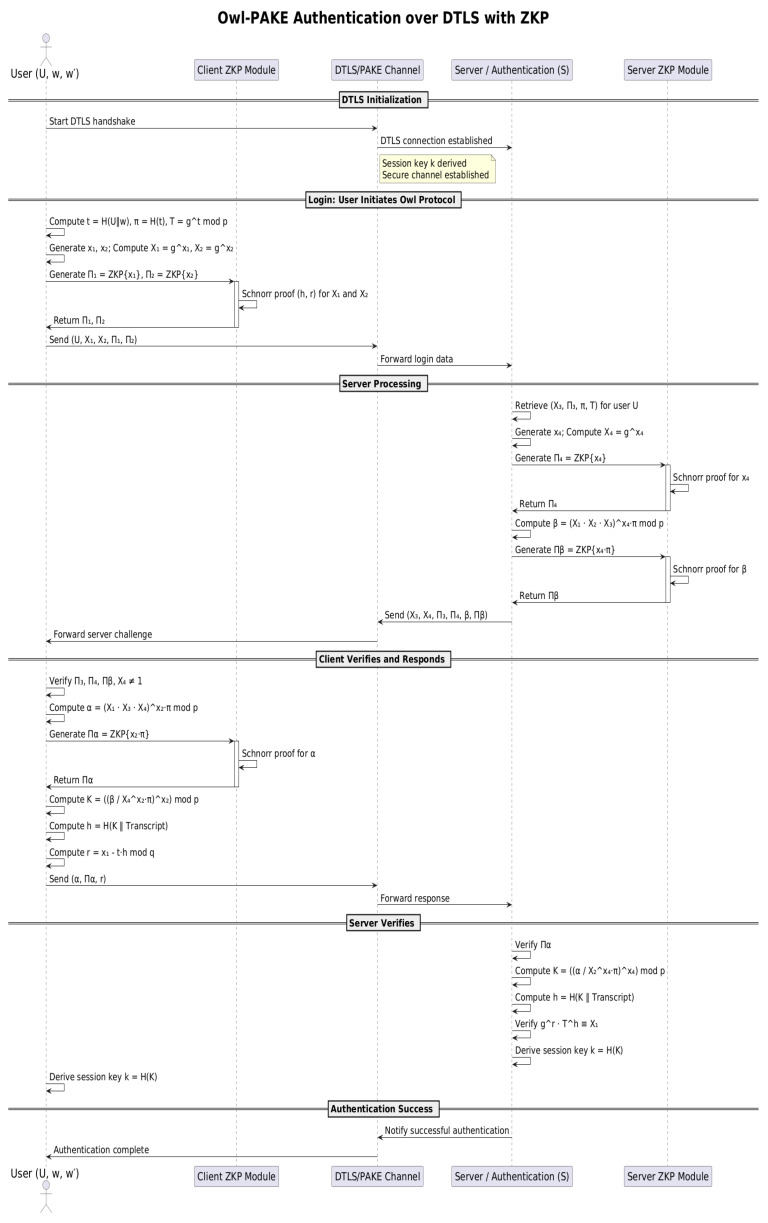
Secure login process in the OWL protocol using ZKPs.

**Figure 3 sensors-25-02468-f003:**
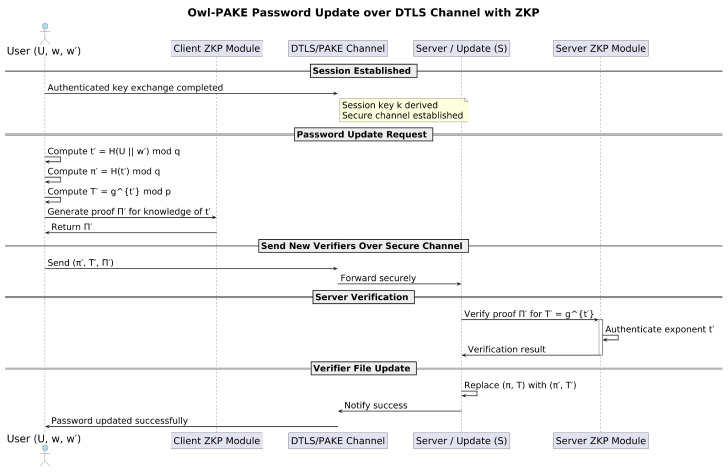
Password update in the OWL protocol using ZKPs.

**Figure 4 sensors-25-02468-f004:**
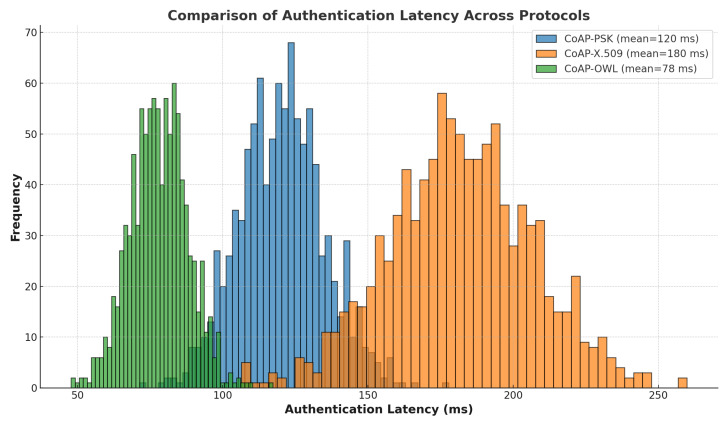
Comparison of authentication latency across protocols.

**Figure 5 sensors-25-02468-f005:**
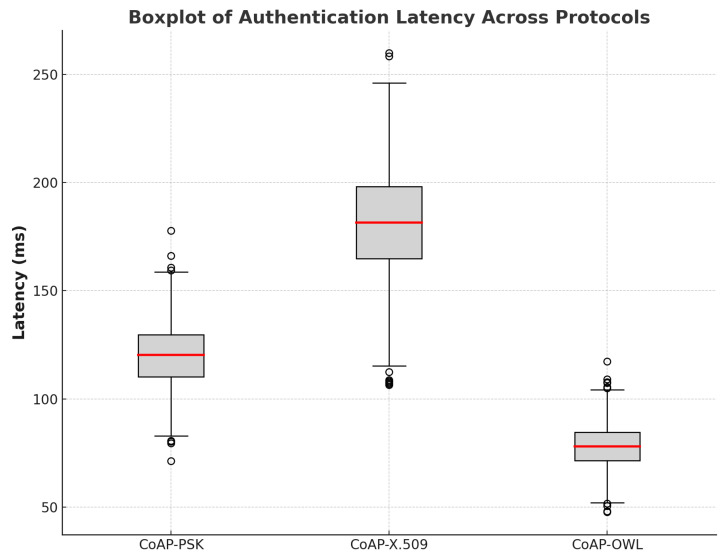
Boxplot of authentication latency across protocols.

**Figure 6 sensors-25-02468-f006:**
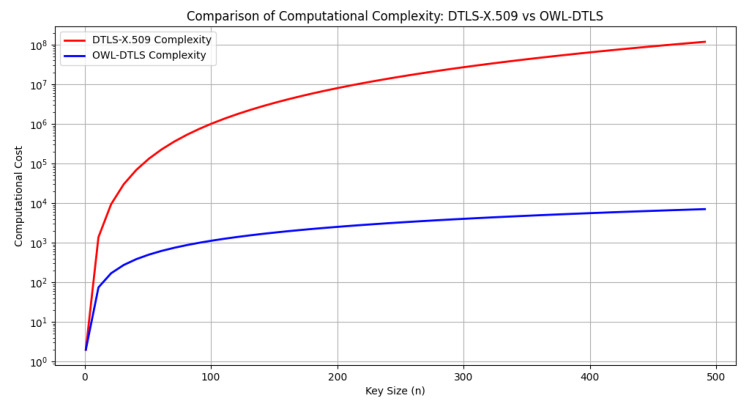
Comparison of computational complexity: DTLS–X.509 vs OWL–DTLS.

**Figure 7 sensors-25-02468-f007:**
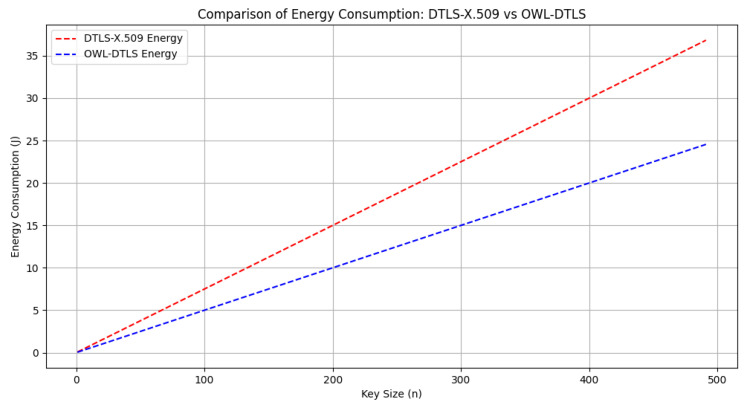
Comparison of Energy Consumption: DTLS-X.509 vs. OWL-DTLS.

**Figure 8 sensors-25-02468-f008:**
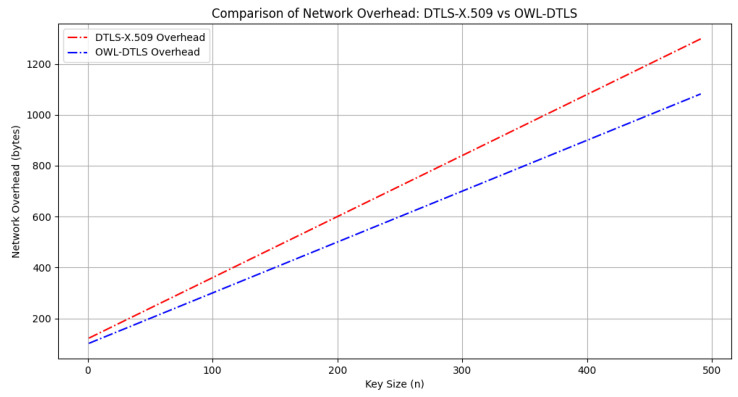
Comparison of Network Overhead: DTLS–X.509 vs. OWL–DTLS.

**Table 1 sensors-25-02468-t001:** Comparison of lightweight PAKE protocols for resource-constrained devices.

Protocol	Year	Innovation	Advantages	Disadvantages	ECC Compatibility	Scalability	Computational Efficiency
EKE (Encrypted Key Exchange) [14]	1992	First formalized PAKE.	Basis for other PAKE protocols. Resists passive dictionary attacks.	Vulnerable to active dictionary attacks. Requires public encryption.	No	Low	Low
SPEKE (Simple PAKE) [15]	1996	Uses modular exponentiation with password hash.	No pre-shared public keys required. Uses hash functions for security.	Vulnerable if parameters are poorly chosen.	No	Medium	Medium
SRP (Secure Remote Password) [16]	1998	Introduces password verification without transmission.	Protects against active and passive dictionary attacks. No plaintext password storage on the server.	More complex to implement. May be less efficient on resource-limited devices.	No	Medium	Low
AugPAKE (Augmented PAKE) [17]	2005	Adds protection for user credentials.	Server does not store plaintext passwords. Resists passive dictionary attacks.	Higher computational overhead than SPEKE and SRP.	No	Medium	Low
J-PAKE (Joy PAKE) [18]	2008	Uses Diffie-Hellman with ZKPs.	No key storage required on the server. Offers security against MitM attacks.	Requires more computational steps than other PAKE protocols.	Yes	High	Medium
Dragonfly [19]	2008	Designed for wireless networks and WPA3.	High resistance to dictionary attacks. No need to store secrets on the server.	May be less efficient on resource-limited devices.	Yes	Medium	Medium
SPAKE2 (Simplified PAKE 2) [20]	2010	Reduces the number of required exponentiations.	More efficient than J-PAKE. Compatible with ECC.	Requires secure parameter generation.	Yes	High	High
OWL (One-Message Weak Leakage-Resilient PAKE) [21]	2015	One-round PAKE with leakage protection.	Reduces latency with a single message. Resists dictionary and replay attacks. Supports implementations in various multiplicative groups and elliptic curves.	Requires stronger security in parameter selection.	Yes	High	High
OPAQUE (Asymmetric PAKE) [22]	2018	No password exposure at any stage.	Supports secure storage using KDF. Provides strong authentication without revealing secrets.	More computationally complex than other PAKEs. Requires encrypted credential storage.	Yes	High	Medium

**Table 2 sensors-25-02468-t002:** Comparison of security protocols in IoT.

Protocol	Security	Efficiency	Compatibility	Communication	Key Exchange	Scalability	Complexity	Performance in Hostile Environments
MQTT [23]	Basic Authentication, Optional TLS	High with Low Bandwidth	Wide on Devices and Platforms	Message-Oriented	No	High	Low	Moderate
CoAP [24]	DTLS, Secure Authentication	High in Low-Power Networks	Specific for IoT Devices	Message-Oriented and RESTful	Yes	Medium	Moderate	High
Zigbee [25]	AES-128 Encryption	Moderate, Optimized for Sensor Networks	Low-Power Devices	Mesh Network	No	High	Moderate	High
BLE [26]	AES Encryption, ECDH for Pairing	High in Personal Devices	Common in Mobile and Health Devices	Point-to-Point Connection	Yes	Low	Moderate	Low
LwM2M [27]	DTLS, Secure Authentication	Designed to Manage IoT Devices	Based on CoAP, Good for IoT Devices	Object Model-Oriented	Yes	High	Moderate	High
LoRa [28]	AES Encryption	High in Long Distances and Low Power	Used in Wide Area Networks (LPWAN)	Mesh Network-Oriented	No	Very High	Low	High
Thread [29]	AES Encryption, IP-Based Security (6LoWPAN)	High in Low-Power Networks	Wide in Smart Home Applications	Mesh Network, IPv6-Based	Yes	High	Moderate	High
Sigfox [30]	Proprietary Encryption	High for Small Messages	Optimized for Low-Power IoT Devices	Uni-Directional or Limited Bi-Directional Communication	No	Very High	Low	High
NB-IoT [31]	3GPP Standard Encryption	High for Small Data Volumes	Integration with Existing Cellular Infrastructure	Narrow Band for IoT	Yes	High	Moderate	Moderate
LTE-M [32]	3GPP Standard Encryption	High with Capacity for Larger Data Volumes	Compatibility with Existing LTE Networks	Suitable for Mobile IoT Applications	Yes	High	Moderate	Moderate
Z-Wave [33]	AES-128 Encryption, ECDH for Pairing	Moderate, Optimized for Smart Home Networks	Preferred in Home Automation	Short-Range Mesh Communication	Yes	Medium	Moderate	High
Wi-SUN [34]	AES-128 Encryption	High in Large-Scale Mesh Networks	Suitable for Public Service Applications	Mesh-Oriented with IPv6	No	High	Moderate	High
ISA100.11a [35]	AES-128 Encryption	High in Industrial Environments	Specific for Industrial Applications	Industrial Wireless Communication	Yes	Medium	Moderate	High
WirelessHART [36]	AES-128 Encryption	Moderate, Specific for Instrumentation	Usage in Process Control Environments	Robust Communication for Process Control	Yes	Medium	Moderate	High
NFC [37]	Standards-based Encryption	High for Short-Range Transmission	Common in Mobile Devices and Payments	Near-Field Communication	Yes	Low	Low	Moderate

**Table 3 sensors-25-02468-t003:** Main components of the OWL–CoAP integration.

Component	Description
**Key Provisioning Module**	Generates and distributes identifiers (e.g., RFC6920-based) for device authentication. Manages secure Access Control Lists (ACLs) with fine-grained permissions. Ensures flexible formats (binary, URI, human-readable) for identifier representation.
**OWL–CoAP Session Manager**	Implements OWL for robust key exchange and mutual authentication. Uses ZKPs to avoid exposing secret credentials. Facilitates session key derivation from password-based inputs.
**CoAP Protocol Adapter**	Translates OWL messages into CoAP-compatible formats (URIs, tokens). Simplifies integration with existing IoT deployments via modular design.
**DTLS Security Layer**	Enables lightweight encryption (PSK) or more advanced RawPublicKey/X.509 setups. Protects data integrity and confidentiality over UDP-based CoAP messages.

**Table 4 sensors-25-02468-t004:** Experimental metrics, factors, and levels.

Metric	Factors	Levels
Computational Efficiency	Key Size, Encryption Method	1–500 bits, OWL-DTLS, DTLS-X.509
Communication Overhead	Protocol Type, Message Size	CoAP-OWL, CoAP-PSK, CoAP-X.509; Small, Medium, Large
Authentication Latency	Authentication Mechanism, Network Conditions	PSK, X.509, OWL; Low, Medium, High Latency
Security Robustness	Attack Type, Encryption Strength	Replay, MITM, Brute Force; Low, Medium, High

**Table 5 sensors-25-02468-t005:** Authentication latency statistics.

Protocol	Mean Latency (ms)	Standard Deviation (ms)
CoAP-PSK	120.29	14.68
CoAP-X.509	181.77	24.92
CoAP-OWL	78.06	9.83

**Table 6 sensors-25-02468-t006:** Detailed complexity comparison between DTLS–X.509 and OWL–DTLS.

Factor	DTLS-X.509	OWL-DTLS	Advantage of OWL
**Key Generation**	$O(n^{3})$	O(nlogn)	OWL reduces the computational cost by using password-derived keys instead of complex asymmetric cryptographic operations.
**Key Verification**	O(nlogn)	O(n)	OWL enables faster key validation without the need for certificate verification.
**Handshake Complexity**	$O(2^{\left(n/50\right)})$	O(n)	OWL simplifies session establishment by reducing handshake steps.
**Encryption Overhead**	O(nlogn)	O(nlogn)	Similar efficiency, but OWL avoids certificate-related overhead.
**Attack Resilience**	Vulnerable to impersonation attacks.	Resistant via ZKPs.	OWL enhances authentication security and prevents identity theft.
**Computational Efficiency**	High CPU/memory usage due to RSA/ECC.	Lightweight cryptographic operations.	OWL is optimized for constrained IoT devices.
**Energy Consumption**	High due to certificate validation.	Low due to minimal computations.	OWL extends battery life in IoT devices.
**Network Overhead**	Large certificate exchanges.	Compact authentication messages.	OWL reduces bandwidth usage, ideal for IoT.

**Table 7 sensors-25-02468-t007:** Sample results for DTLS–X.509 and OWL–DTLS.

Key Size (Bits)	Computational Cost	Energy (J)	Overhead (Bytes)
**DTLS-X.509**			
1	2.00	0.075	122.4
11	1384.75	0.825	146.4
21	9389.87	1.575	170.4
**OWL-DTLS**			
1	2.00	0.050	102
11	74.75	0.550	122
21	169.86	1.050	142

## Data Availability

Data are contained within the article.

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
