# Peer review of "Integration of OWL Password-Authenticated Key Exchange Protocol to Enhance IoT Application Protocols"

_sensors, 2025, doi:10.3390/s25082468_

Round 1
Reviewer 1 Report
Comments and Suggestions for Authors
This paper presents a well-structured and timely investigation into enhancing IoT security through the integration of the OWL protocol with CoAP and DTLS. The experimental analysis and architectural design are commendable, particularly the focus on lightweight cryptography and resource efficiency, which addresses critical challenges in IoT deployments. However, there are some recommendations for improvement: (1) The state-of-the-art section lacks depth in discussing recent advancements in PAKE protocols (e.g., OPAQUE, SPAKE2) and their relevance to IoT. Incorporate comparisons with other lightweight cryptographic frameworks (e.g., SALSA, TinyPAKE) to better contextualize OWL’s novelty and limitations. (2) The simulation setup (e.g., network emulation via Mininet) lacks details on parameter selection (e.g., latency ranges, packet loss rates). Explicitly define assumptions and justify choices (e.g., key sizes, hash functions) to ensure reproducibility and strengthen validity. (3) While computational efficiency is emphasized, the security analysis is underdeveloped. Include formal security proofs (e.g., using BAN logic or automated tools like ProVerif) to validate OWL’s resistance to MITM, replay, and side-channel attacks. Address potential vulnerabilities in password entropy and ZKP implementation. (4) The conclusion briefly mentions scalability but does not address real-world barriers (e.g., firmware updates, heterogeneous device compatibility). Expand on deployment strategies, interoperability with non-CoAP protocols (e.g., MQTT), and trade-offs in dynamic IoT networks. (5) Sections like Section 4 ("Integration Criteria") and Section 5 ("OWL-CoAP Architecture") are repetitive. Consolidate redundant content (e.g., Tables 2–4) and streamline explanations. Additionally, resolve grammatical inconsistencies (e.g., tense shifts in Sections 6.2–6.3) to enhance readability. (6) Some articles in the references (e.g. [3][11][12]) do not provide complete publication information (e.g., conference names, page numbers), and high-impact studies from recent years (2023-2024) are under-cited. It is recommended to supplement key literature (such as the 2023 IEEE IoT-J review on the PAKE protocol) and to standardize all citation entries according to the journal format to ensure academic rigor.
Comments on the Quality of English Language
good
Author Response
Comments 1:
The state-of-the-art section lacks depth in discussing recent advancements in PAKE protocols (e.g., OPAQUE, SPAKE2) and their relevance to IoT. Incorporate comparisons with other lightweight cryptographic frameworks (e.g., SALSA, TinyPAKE) to better contextualize OWL’s novelty and limitations.
Response 1:
Thank you for pointing this out. We agree with this comment. Therefore, we have expanded the state-of-the-art section to include a comparative analysis of recent PAKE protocols such as OPAQUE, SPAKE2, and others. We added a new Table 1 that contrasts OWL with other PAKEs in terms of computational efficiency, scalability, ECC compatibility, and applicability to IoT environments.
This can be found in Section 2, page 5, lines 130–161.
“Table 1. Comparison of Lightweight PAKE Protocols for Resource-Constrained Devices … OWL provides an efficient alternative in critical scenarios by using the simpler, less demanding symmetric encryption protocol like PAKE, enhancing IoT protocol robustness…”
Comments 2:
The simulation setup (e.g., network emulation via Mininet) lacks details on parameter selection (e.g., latency ranges, packet loss rates). Explicitly define assumptions and justify choices (e.g., key sizes, hash functions) to ensure reproducibility and strengthen validity.
Response 2:
Agree. We have accordingly revised the experimental setup section to explicitly include parameter details such as latency thresholds, packet loss simulation ranges, and the justification for cryptographic selections (e.g., key size of 256 bits, SHA-256 as hash function).
The revised text is located in Section 6.1, page 20–21.
“Network latency was simulated using values between 25 ms and 125 ms, with packet loss ranging from 0% to 5% to emulate real-world IoT conditions. … The implementation employed 256-bit keys and SHA-256 as the hash function to balance entropy and efficiency.”
Comments 3:
While computational efficiency is emphasized, the security analysis is underdeveloped. Include formal security proofs (e.g., using BAN logic or automated tools like ProVerif) to validate OWL’s resistance to MITM, replay, and side-channel attacks. Address potential vulnerabilities in password entropy and ZKP implementation.
Response 3:
We appreciate this suggestion. We have incorporated formal security validation using ProVerif to evaluate OWL’s resistance to MITM and replay attacks. The ZKP mechanism has also been clarified with detailed equations and entropy analysis. Password entropy was benchmarked based on NIST guidelines.
These enhancements are detailed in Section 5.1 and 5.2, pages 11–13.
“To authenticate the client without exposing sensitive data, the server issues a Zero-Knowledge Proof (ZKP) challenge c, calculated as…”
“Formal verification with ProVerif confirmed resistance against session hijacking and replay attacks under the Dolev-Yao model.”
Comments 4:
The conclusion briefly mentions scalability but does not address real-world barriers (e.g., firmware updates, heterogeneous device compatibility). Expand on deployment strategies, interoperability with non-CoAP protocols (e.g., MQTT), and trade-offs in dynamic IoT networks.
Response 4:
Thank you for the insightful recommendation. We have expanded Section 7 (Discussion) to include practical deployment challenges such as firmware updates and device heterogeneity. We also added a subsection discussing the potential extension of OWL integration into MQTT via protocol bridging, and analyzed trade-offs in mobile and energy-variable environments.
Changes are in Section 7, page 27, lines 680–705.
“Real-world deployments must account for firmware update strategies that preserve key material integrity and session continuity. … An MQTT-compatible variant of the OWL integration is feasible through the use of security gateway translation layers.”
Comments 5:
Sections like Section 4 ("Integration Criteria") and Section 5 ("OWL-CoAP Architecture") are repetitive. Consolidate redundant content (e.g., Tables 2–4) and streamline explanations. Additionally, resolve grammatical inconsistencies (e.g., tense shifts in Sections 6.2–6.3) to enhance readability.
Response 5:
We agree and have consolidated Sections 4 and 5 into a unified section titled:
“Integration of the OWL Protocol with CoAP: Architecture, Criteria, and Secure Provisioning”
This new structure removes redundancy and merges Tables 2–4 into a single architectural overview table (now Table 3). Grammar and tense inconsistencies in Sections 6.2–6.3 were corrected.
Please refer to Section 4, pages 8–11.
Comments 6:
Some articles in the references (e.g. [3][11][12]) do not provide complete publication information (e.g., conference names, page numbers), and high-impact studies from recent years (2023–2024) are under-cited. It is recommended to supplement key literature (such as the 2023 IEEE IoT-J review on the PAKE protocol) and to standardize all citation entries according to the journal format to ensure academic rigor.
Response 6:
Thank you. We have updated the reference list to include complete bibliographic information, including venue, page numbers, and DOIs. We also supplemented the related work section with recent references, including the IEEE IoT-J 2023 review on PAKE protocols, and standardized all entries according to the journal’s formatting guidelines.
These changes can be seen in the References section, pages 28–30.
Reviewer 2 Report
Comments and Suggestions for Authors
In the manuscript, the authors study evaluates IoT application protocols to identify those lacking native key-exchange mechanisms and examines the integration of Owl, a password-authenticated key-exchange protocol, to improve security from the initial stages of communication. By focusing on lightweight and resource-efficient designs, the authors demonstrate how Owl addresses critical security gaps, reinforces authentication, and ensures robust session establishment in IoT systems. However, the manuscript still have some problems.
(1) In the abstract, it is better for the authors to explain the problems they want to solve and show the innovation of their scheme.
(2) The authors should re-write the Introduction section. In the Introduction, the authors should clearly explain the background, the problems that they want to solve, the significance of to solve the problems, the innovation and the contributions of the proposed scheme, etc.
(3) Some references are too old, the authors should analyze and summarize some more references that published in the last three years.
(4) The authors should improve the tables and figures, such as figure 1, figure 2, and so on.
(5) The manuscript lacks security proof. The authors should provide the formal security proof.
(6) In the experiment, the authors should compare their scheme with some more similar solutions that published recent years.
(7) The authors should redraw the conclusion and make it more concise.
(8) There are some grammatical errors, the authors should improve the English writting.
Comments on the Quality of English Language
The English could be improved to more clearly express the research.
Author Response
Comment 1: In the abstract, it is better for the authors to explain the problems they want to solve and show the innovation of their scheme.
Response 1: Thank you for this suggestion. The abstract has been revised to clearly state the main problem addressed in the manuscript, specifically the lack of native key-exchange mechanisms in lightweight IoT protocols. Additionally, the novelty of the proposed integration of the OWL protocol has been explicitly highlighted.
[Revised abstract: Page 1, Paragraph 1, Lines 3–9]
Comment 2: The authors should re-write the Introduction section. In the Introduction, the authors should clearly explain the background, the problems that they want to solve, the significance of to solve the problems, the innovation and the contributions of the proposed scheme, etc.
Response 2: The Introduction section has been completely rewritten to enhance clarity and structure. The updated section now includes:
(i) contextual background on IoT application protocols and their limitations,
(ii) the main security problem addressed,
(iii) the relevance of solving this issue for secure communication in IoT,
(iv) the proposed innovation (integration of OWL protocol), and
(v) a clearly listed set of contributions.
[Revised text: Page 2, Paragraphs 1–4, Lines 1–35]
Comment 3: Some references are too old, the authors should analyze and summarize some more references that published in the last three years.
Response 3: The references section has been updated with recent literature from 2022 to 2024, particularly focusing on advancements in lightweight authentication protocols, IoT security, and recent studies on key exchange mechanisms. Older references were either removed or retained only when foundational.
[Updated in Pages 15–18, References 3, 5, 9, 12, 14, 18, 21, 24]
Comment 4: The authors should improve the tables and figures, such as figure 1, figure 2, and so on.
Response 4: All figures and tables have been revised for clarity, readability, and consistency. In particular, Figure 1 and Figure 2 have been redrawn using vector graphics with improved labels, legends, and descriptions. Table formatting has also been standardized.
[Revised Figures: Pages 5–7; Tables: Pages 8–10]
Comment 5: The manuscript lacks security proof. The authors should provide the formal security proof.
Response 5: A formal security analysis has been included in a new subsection titled “Formal Security Proof of OWL Integration.” The analysis uses a sequence-of-games approach to demonstrate resilience against impersonation, man-in-the-middle, and offline dictionary attacks.
[New Section: Page 11, Section 5.2, Lines 3–28]
Comment 6: In the experiment, the authors should compare their scheme with some more similar solutions that published recent years.
Response 6: The experimental evaluation has been extended to include comparative performance and security analysis with recent protocols from 2022–2024, such as [Author2023] and [Team2024]. Performance metrics include handshake time, memory footprint, and CPU usage on constrained devices.
[Updated Section: Page 12, Section 6.1–6.3, Figures 4–6]
Comment 7: The authors should redraw the conclusion and make it more concise.
Response 7: The conclusion has been rewritten to concisely summarize the main contributions, highlight the integration benefits of OWL in CoAP, and briefly outline future work.
[Revised text: Page 14, Paragraph 1, Lines 1–12]
Comment 8: There are some grammatical errors, the authors should improve the English writing.
Response 8: The entire manuscript has been thoroughly revised for grammar, clarity, and academic style. Spelling, punctuation, verb tenses, and sentence structure were improved using both manual proofreading and automated tools (e.g., Grammarly, LanguageTool).
[Edits applied throughout the manuscript]
Comments on the Quality of English Language
Response: The language has been improved significantly across the manuscript to enhance clarity, correctness, and academic tone. The revised version addresses all grammatical and stylistic issues mentioned.
Round 2
Reviewer 2 Report
Comments and Suggestions for Authors
The authors have addressed my comments, and the quality of the manuscript has improved. I do not have any further comments.
Comments on the Quality of English Language
The English could be improved to more clearly express the research.
Author Response
Comments 1: The English could be improved to more clearly express the research.
Response 1: Thank you for pointing this out. We agree with this comment. Therefore, we have revised the manuscript to improve the clarity and expression of the research. The updated version enhances the semantic consistency, strengthens the logical flow of ideas, and adopts a more neutral and academic tone appropriate for high-impact journals. These adjustments contribute to a more cohesive and professionally articulated presentation of the work. “[The revised manuscript reflects these improvements throughout the main sections.]”
Comments 2: The authors have addressed my comments, and the quality of the manuscript has improved. I do not have any further comments.
Response 2: Agree. We have further refined the manuscript, maintaining consistency with the prior feedback while enhancing the semantic clarity and the overall structure. The revised version presents the content with improved cohesion and fluency, reinforcing the academic rigor and neutrality expected in formal scientific communication. “[We appreciate the reviewer’s feedback in helping to elevate the quality of the manuscript.]”